# Large Language Model Predicting the Performance of Small Organic Molecule Corrosion Inhibitors

## Abstract

Large language models (LLMs) like GPT-4o have shown promise in solving everyday tasks and addressing basic scientific challenges by utilizing extensive pretrained knowledge. In this work, we explore their potential to predict the efficiency of various organic compounds for the inhibition of corrosion of the magnesium alloy ZE41, a material crucial for many industrial applications. Traditional approaches, such as basic neural networks, rely on non-contextual data, often requiring large datasets and significant effort per sample to achieve accurate predictions. They struggle particularly with small datasets, limiting their effectiveness in discovering new corrosion inhibitors. LLMs can contextualize and interpret limited data points by drawing on their vast knowledge, including chemical properties of molecules and their influence on corrosion processes in other materials like iron. By prompting the model with a small dataset, LLMs can provide meaningful predictions without the need for extensive training. Our study demonstrates that LLMs can predict corrosion inhibition outcomes, and reduce the amount of data needed.

## 1 Introduction

In recent years, LLMs have gained significant attention in both public discourse and academic research, evolving from their original purpose in natural language processing to demonstrating remarkable versatility across various domains (Jablonka et al. (2023); Plaat et al. (2024)). Initially designed to handle tasks such as answering general questions, assisting in programming, and performing simple reasoning tasks, LLMs have since been applied to more complex challenges, including regression, prediction, and classification of numerical datasets.

To enhance the quality and accuracy of LLM-generated responses, researchers have developed strategies like prompt engineering and advanced techniques such as chain-of-thought (CoT, Schulhoff et al. (2024)), tree-of-thought (ToT, Yao et al. (2023)), and skeleton of thought (SoT, Ning et al. (2023)). These methods enable LLMs to produce structured and well-reasoned outputs by allowing them to engage in more reflective and self-guided problem-solving processes. This growing recognition of LLMs as powerful tools has spurred their exploration in fields beyond traditional language processing, including chemistry.

The application of LLMs in chemistry, in particular, has emerged as a promising area of research, with efforts focused on how these models can effectively leverage chemical knowledge. Researchers have employed various approaches to enhance LLM performance in this domain. For example, Hatakeyama-Sato et al. (2023) examined GPT-4's ability to process and apply chemical knowledge through prompt-based interactions, identifying both its potential and its limitations in reasoning and knowledge gaps. In addition, Jablonka et al. (2023) reviewed techniques like fine-tuning GPT-2 and GPT-3.5 and connecting GPT-4 with external tools to improve its utility in chemistry-related tasks. Advancing this work, Bran et al. (2024) developed ChemCrow, a tool that integrates simulations, internet access, and chemical APIs to extend LLM capabilities in molecular transformations and property predictions. Moreover, Jablonka et al. (2024) demonstrated the potential of fine-tuned GPT-3 models in tasks such as classification, regression, and molecule design, while Zhang et al.

(2024) trained a domain-specific LLM, ChemLLM, on a large chemical database, showcasing the value of specialized models in chemical research.

Magnesium (Mg) and its alloys are increasingly used in various novel industrial applications due to their abundance, affordability, and versatility. However, the high chemical reactivity of magnesium requires domain-specific adjustments of its degradation behavior. In transportation (Dziubińska et al. (2016); Joost & Krajewski (2017)), corrosion must be prevented to avoid critical material failure. In medical applications (e.g. temporary biodegradable stents or bone screws, Santos-Coquillat et al. (2019); Witte et al. (2008)) the degradation rate must be precisely controlled to match varying treatment or patient healing rates. For energy applications like Mg-air primary batteries (Höche et al. (2018)) and secondary, a steady rate of Mg dissolution is essential to ensure constant energy output (Ma et al. (2019)). Several strategies, including alloying and surface coatings, have been developed to regulate the degradation of Mg-based materials (Gray & Luan (2002)). A promising approach involves the use of small organic molecules which have shown significant potential in controlling the dissolution properties of pure Mg and its alloys (Lamaka et al. (2017); Blawert et al. (2006)). The vast chemical diversity of organic modulators is a major advantage, offering nearly limitless possibilities for tailored solutions. The number of available organic compounds is rapidly growing, with 120 million new compounds reported in the last decade alone. Estimates suggest that there could be as many as $10^{63}$ (Kirkpatrick & Ellis (2004)) organic compounds with useful properties, making the chemical space effectively infinite. Recent advancements in automation, robotics, and combinatorial chemistry are enabling the synthesis of larger and more diverse chemical libraries and the integration of computer-assisted synthesis methods further accelerates the growth of available compounds.

However, even with the most sophisticated high-throughput techniques available today (White et al. (2012); Muster et al. (2009); Meeusen et al. (2019)), researchers can only explore a tiny fraction of this space. Fortunately, data-driven computational methods can efficiently search larger regions of chemical space with significantly less time and effort, rendering them invaluable for short-listing molecules with desirable properties for specific applications (Winkler (2017); Fernandez et al. (2016); Chen et al. (2016); Winkler et al. (2014); Segler et al. (2018); Coelho et al. (2022); Würger et al. (2022)). Machine learning models that capture complex quantitative structure-property relationships (QSPR) can predict the properties of yet-to-be-synthesized or tested compounds. However, these models require extensive, reliable, chemically diverse, and balanced training datasets to achieve accurate and generalizable predictions. A synergistic approach combining experimental and computational methods forms a robust foundation for data-driven discovery of dissolution modulators (a pool of chemicals that either accelerates or inhibits the corrosion onset and progression in metallic materials) by predicting their corrosion inhibition efficiencies (IEs) prior to testing. To ensure reliable performance, the training datasets must adequately represent the complexity of the relevant chemical environments, especially when predicting properties for molecules with under-represented features in the data. However, the available training in the field of corrosion inhibitor research is quite limited from a machine learning point of view as a large part of chemicals that are labeled with an IE value are proprietary data and not available to the public domain. Fortunately, LLMs show great potential to mitigate the lack of available training data by utilizing their extensive base knowledge to understand contextual data and enhance model predictions. By drawing on a vast repository of scientific information, LLMs can help bridge gaps in existing datasets, supporting more accurate and reliable predictions of inhibition efficiency without the immediate need for additional experimental data.

In this study, GPT-4o is used to predict the efficiency of various compounds to inhibit corrosion of magnesium. It is provided with exemplary data points and compared with baseline results of a neural network from Schiessler et al. (2023). The large language model (LLM) is tasked to predict with two different approaches. The first approach aims at incorporating additional contextual information in combination with the pretrained knowledge of the LLM to increase the accuracy of the predictions. The second approach focuses on evaluating the value of contextual information combined with the pretrained knowledge alone.

## 2 METHODOLOGY

### 2.1 OVERVIEW

This study aims to predict the corrosion inhibition efficiency of various compounds using an LLM by integrating contextual information, including details of the experimental setup, molecular identities, and structural representations. This methodology is intended to enhance predictive performance beyond the neural network baseline reported in Schiessler et al. (2023), which was considered state-of-the-art at the time of publication. With the emergence of advanced LLMs, particularly GPT-4o, novel approaches have become feasible, offering the potential to outperform traditional neural networks, especially when dealing with limited datasets. Given the small dataset of 75 samples used in this study, LLMs can leverage their extensive knowledge base to compensate for data scarcity and improve prediction accuracy. To ensure a fair and direct comparison, the same train-test split as utilized by the neural network model is employed. A naming convention employed in this paper and explained further in the following splits the input data given to the LLM in two parts. The first part, referred to as "data", is the numerical information that is also used for training the neural network. The second part, termed "context", contains all non-numerical information in the form of strings, that can be supplied to an LLM, but not to a basic neural network.

Two distinct prediction approaches are explored:

- **LLM with data and context:** This approach provides the LLM with both the context (molecule names, molecular structure, and experimental details) and the numerical data the baseline neural network had access to during both training and testing. The aim is to leverage the LLM's inherent knowledge and reasoning capabilities to integrate these data sources and outperform the neural network baseline.

- **LLM with context only:** In this approach, the LLM is provided with only the context information (molecule names, molecular structure, and experimental details), without the numerical data. This setup tests the LLM's ability to generate accurate predictions based purely on its understanding of chemical properties and the provided experimental context.

By integrating this contextual data into the LLM's prompting strategy, we hypothesize that the model can better understand the underlying chemical processes, ultimately leading to more accurate predictions of inhibition efficiency than the neural network model, which lacks access to this qualitative information. Furthermore, we hypothesize, that the context information together with the pretrained knowledge of GPT-4o can replace the numerical descriptor data.

### 2.2 DATA AND BASELINE

The baseline results and input data are sourced from Schiessler et al. (2023). The dataset comprises 75 chemical compounds along with their corresponding inhibition efficiencies, which serve as labels. Inhibition efficiency measures the extent to which a given compound alters the rate of corrosion compared to a control experiment with no compounds present. This value is expressed as a percentage, where 100% indicates a perfect inhibitor that completely halts corrosion. Conversely, negative values suggest that the compound accelerates the corrosion process. In this study, the unit of the inhibition efficiency is denoted as IE to clearly distinguish it from other percentage-based metrics.

The input dataset includes approximately 1,250 molecular descriptors for each compound. The descriptors were generated using the chemoinformatics software package alvaDesc and the quantum chemical software package TurboMole to encode the molecular and in part the electronic structure of the 75 molecules that were used in the benchmark study.

To quantify the corrosion inhibition efficiencies of the used benchmark dataset, hydrogen evolution measurements have been performed following the experimental procedure described in the benchmark study.

The predictive model utilized by Schiessler et al. (2023) is a neural network. The model receives as input five molecular descriptors out of the available 1,250, selected through a recursive feature elimination algorithm. Among the various feature sets examined by Schiessler et al. (2023), this

paper focuses exclusively on the FS60 set. This set is selected using only the training data, ensuring that it remains unbiased by excluding the test data from the feature selection process. The descriptors in FS60 include P_VSA_MR_5, LUMO, E1p, CATS3_02_AP, and Mor04m. Consequently, only the predictions generated using this feature set will be considered for comparison. Furthermore, to maintain consistency, the same train-test split as in the original study will be used for prediction.

## 2.3 PROMPTING STRATEGY

### 2.3.1 TECHNOLOGY

The large language model (LLM) employed in this study is GPT-4o by OpenAI, accessed via Microsoft Azure. The specific version used is "2024-02-15-preview", configured with a temperature of 0.7 and a top_p value of 0.95. Detailed prompting strategies are included in the appendix.

### 2.3.2 CONTEXT AND DATA

"Context" comprises molecule names, SMILES (Simplified Molecular Input Line Entry System) strings, and additional information about the experimental setup, while "Data" refers to the molecular descriptors available to the neural network baseline. SMILES strings offer a concise and standardized representation of molecular structures. These SMILES strings are sourced from the PubChem database (Kim et al. (2022)). For the remaining context information, the LLM is provided with two prompts that define its role as a professional chemist, outline the importance of predicting inhibition efficiency, describe the experimental settings, and detail the task at hand. These prompts encourage the LLM to approach the problem systematically, working step-by-step to ensure a thorough and accurate analysis.

Upon providing GPT-4o with the dataset, the model undergoes an analytical process structured into four distinct prompts. In the first prompt, GPT-4o is instructed to generate a comprehensive list of functional groups and atomic structures for each molecule. The second prompt directs the model to autonomously reduce the molecular descriptors from five to a more manageable set of 2-3 per molecule, as the inclusion of all five descriptors often led to data overload, resulting in decreased prediction accuracy. The third prompt combines the outputs from the first two steps into a single, unified dataset. Finally, the fourth prompt guides GPT-4o to examine the consolidated data, identifying patterns and relationships between the input features and inhibition efficiency labels. This refined analysis, coupled with the optimized dataset, serves as the foundation for subsequent prediction prompts, enhancing the model's overall reliability and precision.

### 2.3.3 PREDICTION

The prediction process involves a guided, step-by-step prompt consisting of multiple steps. Key steps include:

- **Finding similar molecules:** The LLM identifies molecules in the training set that are most similar to the test molecule, using both original data and previously analyzed functional group data.
- **Ranking for similarity:** The LLM assigns similarity scores to these molecules, facilitating a weighted average calculation.
- **Analyzing the test molecule:** The LLM synthesizes and analyzes functional group and atomic structure data to make an educated guess on the inhibition efficiency.
- **Weighted average calculation:** A weighted average of inhibition efficiencies from similar molecules is computed, with a reevaluation of similar molecules beforehand.
- **Reevaluation and correction:** If significant variability is found, the LLM employs a more sophisticated approach trying to analyze the corrosion mechanisms of similar molecules.

### 2.3.4 REPETITIONS AND STRATEGY

Due to the non-deterministic output of the LLM, the prediction process is repeated 20 times, with the mean result used as the final output. Lowering the temperature setting to reduce randomness resulted in poorer predictions.

In the "LLM with context only" experiment, molecular descriptors are excluded, requiring the LLM to conduct a more thorough analysis of functional groups and atomic structures. The data analysis steps are consolidated, and the focus shifts to leveraging the model's internal chemical knowledge to identify patterns and predict inhibition efficiency.

## 3 RESULTS

### 3.1 PERFORMANCE METRICS OVERVIEW

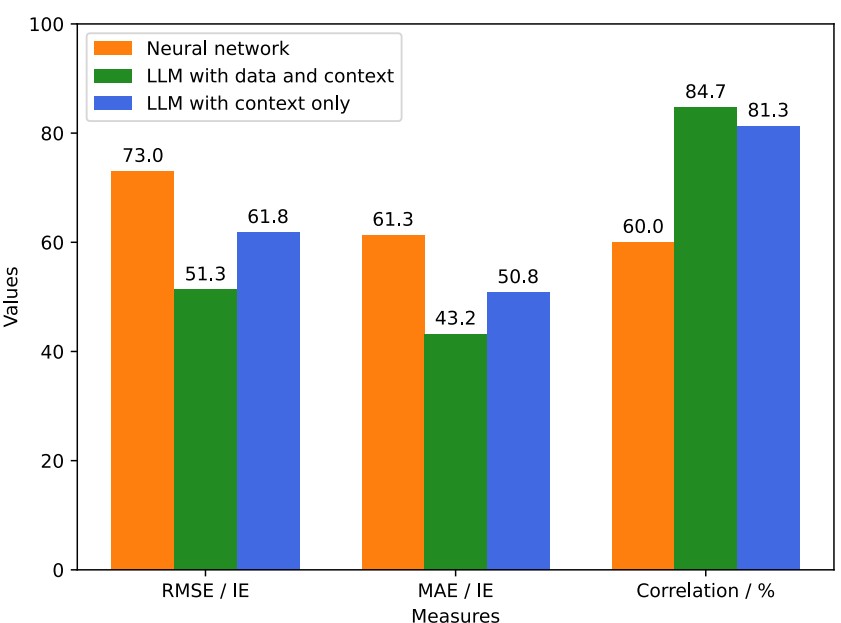

Figure 1: Overview of the results using different evaluation metrics. The neural network results are sourced from Schiessler et al. (2023). The correlation metric indicates the relationship between the predictions and the experimental values, where higher values signify better results. Conversely, lower values for the error metrics RMSE (root mean squared error) and MAE (mean absolute error) denote improved performance.

Figure 1 presents three evaluation metrics: root mean squared error (RMSE), mean absolute error (MAE), and the correlation between the predictions and the experimental results. It is evident that GPT-4o outperforms the neural network in both configurations across all metrics. A higher correlation value indicates superior predictive performance, while smaller RMSE and MAE values correspond to better accuracy.

Comparisons of RMSE and MAE values reveal that the LLM with both data and context yields the most accurate predictions, as indicated by significantly lower error measures. This suggests that the incorporation of contextual information, in conjunction with the LLM's inherent knowledge, enhances prediction quality. Additionally, the second experiment, which excludes the descriptor data, also demonstrates improved precision, albeit to a lesser extent. This implies, that the contextual information, when combined with GPT-4o, result in more accurate predictions than those produced by the neural network using the numerical descriptor dataset from Schiessler et al. (2023) alone.

Examining the correlation values provides further insights. Although the order of the values remains consistent, the difference between the two LLM approaches is notably smaller. The neural network results exhibit a moderate correlation between the predictions and the experimental values (0.6). Both LLM approaches show similar correlation values of 0.847 and 0.813, respectively. The comparable correlation values, along with the observed discrepancies in prediction errors, indicate

that there exists an appropriate linear transformation that could significantly reduce the errors of the LLM without data. In other words, the predictions exhibit systematic errors that could be removed.

## 3.2 Prediction Accuracy Analysis

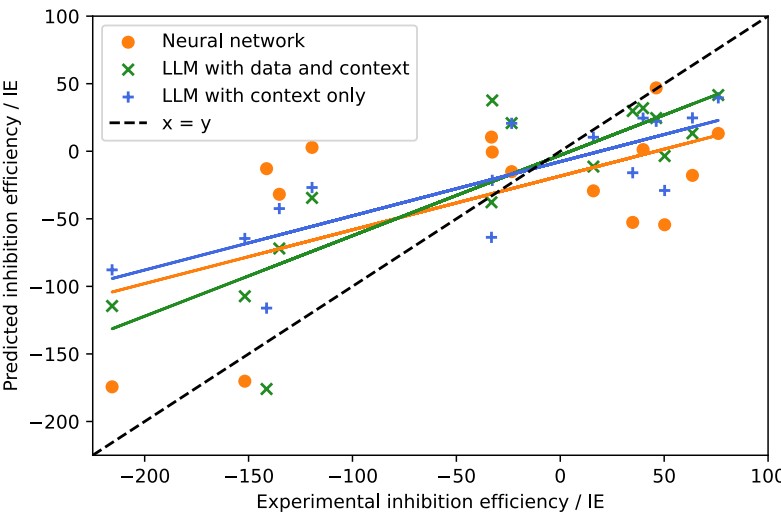

Figure 2: Comparison of LLM predictions and baseline results from Schiessler et al. (2023), shown pointwise. The straight lines represent linear fits of the data points, with predicted values plotted on the y-axis against the experimental values on the x-axis. The dashed line indicates the hypothetical optimal fit. Notably, the LLM data points align more closely with their fit line compared to the neural network results.

In Figure 2, the prediction results are presented for each test sample and prediction method. For each method, a linear curve fit is applied to the results. It can be observed that the linear fit lines do not differ as markedly as the results in Figure 1. The differences in results are attributed to the variance in the locations of individual points around their respective fit lines. The neural network results exhibit the largest variance around their fit line, whereas the LLM prediction results show significantly lower variance. For example, aside from the two leftmost points of the neural network, the other points appear to be randomly scattered around a constant value. In contrast, the LLM predictions demonstrate a clear correlation between predicted and experimental values, consistent with the correlation values shown in Figure 1.

Comparing the distance of the points to the ideal line (black dashed line, $x = y$), it is evident that the overall distance of the LLM results is smaller than that of the neural network results. Notably, the results of the LLM with both data and context are closer to the optimal results. Furthermore, the curve fit of these results is the closest approximation to the optimal results line.

One notable observation is the tendency to underestimate positive inhibition efficiencies and overestimate negative ones. This tendency was also reported by Schiessler et al. (2023). Additionally, the LLMs exhibit a slight bias towards higher inhibition efficiencies. The observed biases are approximately 2.8 IE for the neural network, and around 13.3 IE and 11.2 IE for the LLMs with and without data, respectively. During testing, it was found that the LLM results initially had even higher biases, which were reduced after providing access to the entire test set before prediction. Additionally, clarifying that both inhibition efficiencies and acceleration efficiencies (with negative values) were being sought helped in reducing the bias.

Overall, the LLM approaches, especially when provided with both data and context, demonstrate superior performance in predicting corrosion inhibition efficiencies compared to the traditional neural network approach. This underlines the potential of LLMs in leveraging contextual information and their inherent chemical knowledge to achieve more accurate predictions in the field of materials science.

## 3.3 DISTRIBUTION OF PREDICTION RESULTS

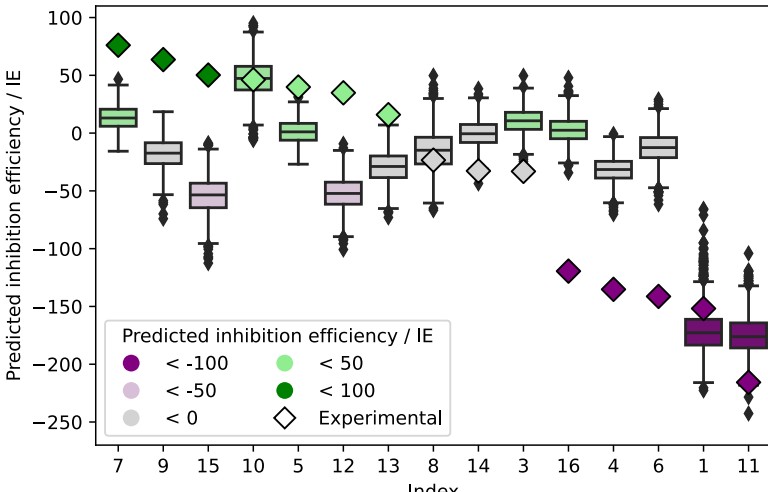

Figure 3: Baseline: Adapted figure from Schiessler et al. (2023). The distribution of the predicted inhibition efficiencies along with their experimental values. Indices are ranged from 1 to 16, excluding 2 (15 total). 1000 predictions were made for each sample.

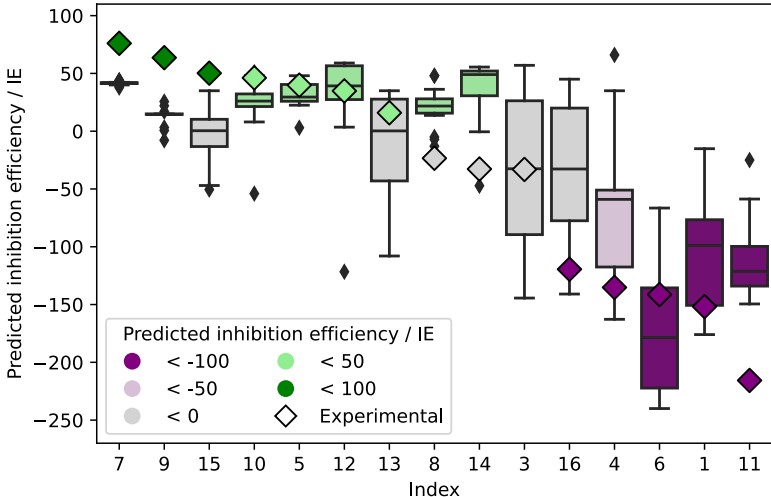

Figure 4: Distribution of the results from the LLM with data and context. Samples are sorted as in the previous figure. 20 predictions were made for each sample.

Figures 3, 4, and 5 illustrate the distributions of prediction results across different approaches. The neural network predictions, as depicted in Figure 3 (adapted from Schiessler et al. (2023)), show relatively constant spreads across all samples. This spread is partially attributed to the varying validation sets employed during training by Schiessler et al. (2023). In contrast, predictions using GPT-4o were generated without employing a validation set, and only 20 repetitions were performed instead of 1000 due to constraints related to computational time and cost.

In contrast, Figure 4 presents the results from the LLM approach, which incorporates both data and context. Here, the prediction spreads vary significantly across different samples. For example, the spread for sample 7 is nearly 0 IE, while for sample 3, it reaches approximately 200 IE. Additionally,

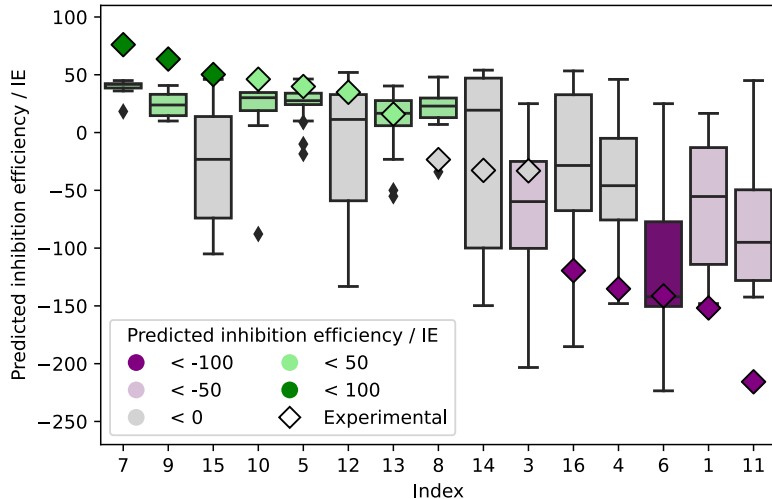

Figure 5: Distribution of the results from the LLM with context only. Samples are sorted as in the previous figure. 20 predictions were done for each sample.

the distributions of the LLM results exhibit considerable inhomogeneity. Unlike the neural network results, where the quantiles are nearly symmetric around the median, the LLM results display a more irregular pattern. Although the LLM predictions are based on only 20 iterations per sample, the trend towards higher inconsistency is evident in comparison to the neural network (1000 repetitions).

In Figure 5, it is apparent that the spread of prediction results has increased for most samples compared to Figure 4. Notably, samples 15, 14, and 6 demonstrate significantly larger spreads, with sample 6 exhibiting the widest range of results, approximately 250 IE. In contrast, sample 13 displays a reduced variance in Figure 5, highlighting a deviation from this general trend.

Furthermore, the variance in LLM predictions is particularly pronounced for accelerators (molecules predicted as accelerators), while it is generally smaller for inhibitors. Analysis of the LLM's outputs suggests that GPT-4o possesses more extensive knowledge about inhibition mechanisms than acceleration mechanisms, which may contribute to this outcome. The term "inhibition efficiency" might also influence GPT-4o's analysis, potentially biasing the model's interpretation toward inhibition. This bias had been observed in the previous section. Another contributing factor could be the lower density of accelerator samples in the dataset.

Overall, the fluctuations in GPT-4o's predictions can be primarily attributed to its reliance on analyzing similar molecules. The consistency in selecting similar molecules across predictions suggests that GPT-4o has developed a coherent understanding of molecular similarity. However, when the inhibition efficiencies of these similar molecules vary widely, the resulting predictions can fluctuate within the same range. In such cases, GPT-4o may base its approximation on different subsets of similar molecules, leading to variability in the outcomes.

The relatively small variance in the neural network results is expected, given the limited variability in the input data. However, the variance observed in the LLM results, especially the differing variances across samples, presents a more complex picture. Likely contributing factors include the number and diversity of similar molecules, as well as how well these molecules were represented in GPT-4o's training set. Occasionally, high variance might also stem from hallucinations, though testing revealed that GPT-4o rarely invented inhibition efficiencies for test molecules, preferring instead to approximate based on known data. The inclusion of descriptor data appears to enhance the consistency of GPT-4o's predictions. Interestingly, the predictions for inhibitors tend to be more stable than those for accelerators, which may be due to the model's more robust understanding of inhibition mechanisms and the higher sample density in that regime.

## 4  DISCUSSION

The small size of the test set with 75 samples raises concerns about the generalizability of the findings. Nevertheless, the neural network exhibited significant difficulties in making accurate predictions. These inaccuracies were mitigated through the use of GPT-4o for prediction. The substitution of the input data with molecular names and SMILES strings, supplemented by contextual information, underscores the utility of GPT-4o's embedded knowledge, as evidenced by a substantial reduction in error. The limited dataset size does not detract from this observation.

For larger datasets, the proposed approach may encounter challenges, particularly in data analysis, which might need to be conducted in batches. Additionally, it remains unclear how well this approach scales with larger datasets and more extensive context windows, as maintaining an overview of the data may become increasingly difficult.

One potential method to reduce underestimation and overestimation errors involves employing a linear mapping, estimated using predictions on a small portion of the training set. Given the high correlation of the LLM results but significant discrepancies in the slopes of the linear fits (Figure 2), this technique could potentially decrease the prediction error.

## 5  CONCLUSION

This study demonstrates the advantages of utilizing a large language model (LLM) like GPT-4o for prediction tasks on small datasets. The predictions generated by GPT-4o, when supplemented with additional contextual information, significantly outperform those from a traditional neural network.

Two experiments were conducted to compare the prediction capabilities of GPT-4o against a neural network. In the first experiment, the LLM received the same data as the neural network, along with a description of the problem setting and contextual information about the samples (SMILES strings of the molecules and their names). The results of this experiment, when compared to the baseline neural network, indicated a substantial reduction in error. The root mean squared error (RMSE) decreased from 73 to 51.3, and the mean absolute error (MAE) decreased from 61.3 to 43.2 (Figure 1). These findings suggest that leveraging the knowledge embedded in GPT-4o, along with a detailed problem description and contextual data, can significantly enhance prediction accuracy.

In the second experiment, the LLM was not provided with the data used by the neural network. Instead, it was given only the problem setting and contextual information. This approach also led to a reduction in prediction error. The RMSE decreased from 73 to 61.8, and the MAE decreased from 61.3 to 50.8 (Figure 1). These results imply that the combined knowledge of GPT-4o, along with a problem description and contextual information, is more valuable for prediction than the molecular descriptor data alone with a neural network.

The outcomes of these experiments illustrate the potential of LLMs to replace neural networks for prediction tasks, particularly when dealing with small datasets. The extensive knowledge base of LLMs makes this approach especially promising for cases where complex dependencies cannot be captured solely by the data. Furthermore, in situations where parts of the data are challenging to input into a neural network, such as strings and text, LLMs may enhance the quality of the results. Another significant advantage is that, for small datasets, there is no need for extensive training. If the data can be provided through prompts, the setup of the prediction process becomes straightforward and does not require substantial computational resources, at least not at the users machine.

### REPRODUCIBILITY

The exact design of the prompts and the order of them being given to GPT-4o is stated in the appendix. With these prompts and the data available on zenodo (from Schiessler et al. (2023)) `https://doi.org/10.5281/zenodo.7780743` the experiments from this paper can be done again. It was tried to generate exactly reproducible results with the seed functionality (`https://learn.microsoft.com/en-us/azure/ai-services/openai/how-to/reproducible-output?tabs=python`), but as stated, the determinism can break for longer responses. For the answer generation for the prompts here, this did break consistently. The best found solution was the repetition of the experiment 20 times and taking the mean over all

results. Trying to reduce the variance of the results with for example a reduced temperature led to worse results.

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

## A PROMPTS

In the following the used prompts are listed. The prompts, after which GPT-4o is tasked to generate an answer, are stated explicitly. It then gets the whole list of previous prompts if not differently stated. The prompts, after which an answer will be generated, will not be appended to the list of previous prompts.

### A.1 EXPERIMENT 1

Role description and motivation (system)

You are a professional chemist with deep knowledge about organic chemistry and corrosion mechanisms. The objective is to predict the corrosion inhibition efficiency of various organic compounds for magnesium (Mg) in salt water. Corrosion inhibition efficiency is critical for preventing the degradation of materials in industrial applications. This dataset contains molecular structures along with their respective inhibition efficiencies expressed in percentages (cannot be larger than 100), negative values express an acceleration of the corrosion process. You are tasked with predicting the inhibition efficiencies of the compounds in the test dataset. do follow the steps in the prompt step by step. think systematically and structured.

Problem description (system):

In the field of corrosion science, corrosion inhibitors are chemical compounds that, when added to the environment in small concentrations, significantly reduce the rate of corrosion. Corrosion accelerators are compounds that increase the rate of corrosion. The effectiveness of an inhibitor depends on its molecular structure and its ability to interact with the metal surface. This problem involves predicting the inhibition efficiency of magnesium (Mg, ZE41) using a set of organic compounds. You will be provided with two datasets: A training dataset with labeled inhibition efficiencies (ie_ze41) to identify patterns and relationships between molecular structures and their corrosion inhibition efficiencies. A test dataset without labels, for which you will predict the inhibition efficiencies based on the patterns learned from the training dataset.

Training data (user):

This is the training data:
names: [name1, name2, ...]
isomeric_smiles: [smiles1, smiles2, ...]
descriptor1: [...]
descriptor2: [...]
...
The inhibition efficiencies of the components are: [IE1, IE2, ...]

Analysis of functional groups (user). After this prompt GPT-4o is tasked to generate, the result is appended to the prompts list after the next prompt (assistant):

write down all functional groups and atomic structures together with their inhibition efficiency for each sample of the training data in a list.

Analysis of descriptors (user). After this prompt GPT-4o is tasked to generate, the result is appended to the prompts list (assistant):

write down the values and names of 2-3 non zero descriptors together with their inhibition efficiency for each sample of the training data in a list.

Combination of data (user). After this prompt GPT-4o is tasked to generate, the result is appended to the prompts list (assistant):

create a list where you combine the functional groups and atomic structures with the non zero descriptors and their inhibition efficiency for each sample of the training data.

Analysis of data (user). After this prompt GPT-4o is tasked to generate, the result is appended to the prompts list (assistant).

search for patterns in the previously created list and analyze the influence of the functional groups, atomic structures and non zero descriptors on the inhibition efficiency.

Overview on the test data (user):

This is the test data:
names: [name1, name2, ...]
smiles: [...]
descriptor1: [...]
...

Prediction prompt (user). This prompt is given to GPT-4o for each test molecule once. The result is not appended to the list of previous prompts.

This is the test data: "Test molecule name, smiles string and data"
Do this step by step:
step 1: Find all similar molecules in the training data and analyze their relation to this compound wrt the magnesium corrosion process. Use the training data and the analyzed training data.
step 2: Analyze similar molecules and rank them by similarity (wrt the mechanisms in the corrosion process). Assign them a similarity value (wrt the mechanisms in the corrosion process).
step 3: Analyze if found patterns apply to the molecule
step 4: Analyze its functional groups and their influnce on the inhibition efficiency
step 5: Analyze its atomic structure and how this might influence the inhibition efficiency
step 6: Make an educated guess for the inhibition/acceleration efficiency of this compound.
step 7: Calculate a weighted average of the inhibition efficiencies of the similar molecules. Exclude molecules that have a small similarity value.
step 8: If the similar molecules have very different inhibition efficiencies, you must make a prediction based on the atomic structure and functional groups as this indicates that the corrosion process is not the same. You must take a closer look on the corrosion mechanisms in that case. Analyze, which molecules might have a similar corrosion mechanism. For that, analyze the corrosion mechanisms for each molecule. After that, decide, which corrosion mechanism is the most likely for the molecule. Use only the molecules which have the same corrosion mechanism.
step 9: Review your analysis shortly and write down your prediction. There were multiple ways predicting the inhibition efficiency. Decide for one way and use only this way.
step 10: As a result, write down one value and nothing after that.

## A.2 EXPERIMENT 2

Role description and motivation (system):

You are a professional chemist with deep knowledge about organic chemistry and corrosion mechanisms. The objective is to predict the corrosion inhibition efficiency of various organic compounds for magnesium (Mg) in salt water. Corrosion inhibition efficiency is critical for preventing the degradation of materials in industrial applications. This dataset contains molecular structures along with their respective inhibition efficiencies expressed in percentages (cannot be larger than 100), negative values express an acceleration of the corrosion process. You are tasked with predicting the inhibition efficiencies of the compounds in the test dataset. do follow the steps in the prompt step by step. think systematically and structured.

Problem description (system):

In the field of corrosion science, corrosion inhibitors are chemical compounds that, when added to the environment in small concentrations, significantly reduce the rate of corrosion. Corrosion accelerators are compounds that increase the rate of corrosion. The effectiveness of an inhibitor depends on its molecular structure and its ability to interact with the metal surface. This problem involves predicting the inhibition efficiency of magnesium (Mg, ZE41) using a set of organic compounds. You will be provided with two datasets: A training dataset with labeled inhibition efficiencies (ie_ze41) to identify patterns and relationships between molecular structures and their corrosion inhibition efficiencies. A test dataset without labels, for which you will predict the inhibition efficiencies based on the patterns learned from the training dataset.

Training data (user):

This is the training data:
names: [name1, name2, ...]
isomeric_smiles: [smiles1, smiles2, ...]
descriptor1: [...]
descriptor2: [...]
...
The inhibition efficiencies of the components are: [IE1, IE2, ...]

Overview on the test data (user):

> This is the test data:
> names: [name1, name2, ...]
> smiles: [...]

Analysis of functional groups (user). After this prompt GPT-4o is tasked to generate, the result is appended to the prompts list (assistant).

> Step 1: Identify all functional groups and other chemical properties you know for each sample of the training data. Analyze their influence on the inhibition efficiency.
> Step 2: Analyze the atomic structures of the compounds and their influence on the inhibition efficiency.
> Step 3: Find compounds in the training data that are similar but have different inhibition efficiencies. List them. Explain, why these differences lead to different inhibition efficiencies. Use a systematic approach and think step by step.

Prediction prompt (user). This prompt is given to GPT-4o for each test molecule once. The result is not appended to the list of previous prompts.

> This is the test data: "Test molecule name and smiles string"
> Do this step by step:
> step 1: Find all similar molecules in the training data and analyze their relation to this compound wrt the magnesium corrosion process. Use the training data and the analyzed training data.
> step 2: Analyze similar molecules and rank them by similarity (wrt the mechanisms in the corrosion process). Assign them a similarity value (wrt the mechanisms in the corrosion process).
> step 3: Analyze if found patterns apply to the molecule
> step 4: Analyze its functional groups and their influnce on the inhibition efficiency
> step 5: Analyze its atomic structure and how this might influence the inhibition efficiency
> step 6: Make an educated guess for the inhibition/acceleration efficiency of this compound.
> step 7: Calculate a weighted average of the inhibition efficiencies of the similar molecules. Exclude molecules that have a small similarity value.
> step 8: If the similar molecules have very different inhibition efficiencies, you must make a prediction based on the atomic structure and functional groups as this indicates that the corrosion process is not the same. You must take a closer look on the corrosion mechanisms in that case. Analyze, which molecules might have a similar corrosion mechanism. For that, analyze the corrosion mechanisms for each molecule. After that, decide, which corrosion mechanism is the most likely for the molecule. Use only the molecules which have the same corrosion mechanism.
> step 9: Review your analysis shortly and write down your prediction. There were multiple ways predicting the inhibition efficiency. Decide for one way and use only this way.
> step 10: As a result, write down one value and nothing after that.

