# OpenReview forum: "Large Language Model Predicting the Performance of Small Organic Molecule Corrosion Inhibitors"
_ICLR.cc/2025/Conference — ICLR 2025 Conference Withdrawn Submission_

### Official Review · Reviewer_ecCv · 2024-11-02

**Soundness:** 2
**Presentation:** 1
**Contribution:** 1
**Rating:** 3
**Confidence:** 4

**Summary:**

This paper aims to use LLMs on the inhibition of ZE41 prediction tasks. The motivation is that the LLMs are better for the few-shot tasks where data are limited. The results shows that the LLMs can have better performance than traditional approaches.

**Strengths:**

Using LLMs on scientific tasks like chemistry are important research topic.

**Weaknesses:**

-	Lack of machine learning insights: The methods used in this paper are standard pipelines in the NLP tasks. There are no special designs for these chemical tasks.
-	Limited experiments: the only model used is GPT-4o and asks are only for ZE41. It’s not clear how these finding will generalize to other settings.
-	Lack of baselines: the only one is neural network, not comparing with other few-shot methods.

**Questions:**

- Can authors provide more results on different LLMs and chemical tasks?

---

> ### Author Response · Authors · 2024-11-18
> **Thank you for the feedback**
>
> Dear Reviewer,
> thank you very much for your review. I understand that our paper is not sufficient for the ICLR and therefore we will withdraw it. We thank you for your feedback.
> Regarding your question:
> We will try that in the future, at least for different LLMs, and, if we find suitable datasets, also for that.

---

### Official Review · Reviewer_eHSq · 2024-11-03

**Soundness:** 1
**Presentation:** 2
**Contribution:** 1
**Rating:** 1
**Confidence:** 4

**Summary:**

This paper proposes an LLM-based prompting approach for alloy corrosion inhibition efficiency prediction of small organic molecules. When comparing with a feature-based multi-layer perceptron baseline, the authors found that the proposed LLM-based approach achieves lower loss and higher correlation coefficients, therefore concluding that the LLM approach outperforms the baseline method.

**Strengths:**

-The authors propose a sophisticated and useful prompting approach based on the OpenAI GPT4o.

-The application of LLM method in corrosion inhibitor prediction is novel as far as I know. and the message conveyed from the results and presentation is clear.

**Weaknesses:**

A significant weakness of this work, in my opinion, is its appropriateness to ICLR audience, or general ML community: the work introduces LLM prompting methods into a new application of corrosion inhibitor prediction, but the insights for LLM for molecular property prediction[1,2] is not novel, neither highlighting its usefulness on small data[3].

Further, given the small size (75) of the studied dataset, in my opinion, the corrosion inhibitor efficiency prediction is not really an interesting ML problem, since sophisticated deep learning approach will not be very useful here. The baseline used in this work confirms that, which is a chemical-descriptor-based multi-layer-perceptron (MLP) model.

The computational experiment is also a bit lacking. The authors only tried GPT-4o model, but does it generalize? Is it actually reproducible? The authors should also add numbers from other leading close-source and open-source models (particularly important for reproducibility concerns, since GPT models can be deprecated). Besides, the author should extend the methods to multiple datasets and prove its generalization power. In terms of baseline, there is only one from an existing work, and it's an MLP-based approach, how about random forest, support vector machines, and other types of chemical features or descriptors? The authors should also consider adding baseline of pretrained chemical foundation models[4-6, these references are a bit outdated, just presented as examples. I encourage the authors to check the SOTA pretrained foundation models for molecular property prediction and do fine-tuning on top of it].

Overall, I will not recommend consideration for acceptance on this work.

[1] https://github.com/ChemFoundationModels/ChemLLMBench; What can Large Language Models do in chemistry? A comprehensive benchmark on eight tasks, NeurIPS 2024.

[2] 14 examples of how LLMs can transform materials science and chemistry: a reflection on a large language model hackathon, Digital Discovery, 2023,2, 1233-1250

[3] Jablonka, K.M., Schwaller, P., Ortega-Guerrero, A. et al. Leveraging large language models for predictive chemistry. Nat Mach Intell 6, 161–169 (2024). https://doi.org/10.1038/s42256-023-00788-1

[4] ChemBERTa: Large-Scale Self-Supervised Pretraining for Molecular Property Prediction. arXiv:2010.09885

[5] Ross, J., Belgodere, B., Chenthamarakshan, V. et al. Large-scale chemical language representations capture molecular structure and properties. Nat Mach Intell 4, 1256–1264 (2022). https://doi.org/10.1038/s42256-022-00580-7

[6] GROVER: https://github.com/tencent-ailab/grover

**Questions:**

The authors present a molecular-similarity-based LLM prompting approach, have they tried a baseline ML approach, non-LLM, using a similar logic? For example, using feature-based similarity to find relevant molecules in the training dataset and then averaging their properties for the IE prediction.

**Details Of Ethics Concerns:**

N.A.

---

> ### Author Response · Authors · 2024-11-18
> **Thank you for the constructive feedback!**
>
> Dear Reviewer,
> thank you very much for your review. I understand that our paper is not sufficient for the ICLR and therefore we will withdraw it. We thank you for your constructive feedback and your suggestions for improvement.
> Regarding your questions:
> The baseline neural net approach is not directly similarity related, but effectively doing exactly that (feature based similarity). The problem is, that we do not have a similarity measure for molecules on that problem, because we do not know enough about the specific mechanisms. The baseline paper is trying to do exactly that by analyzing feature importance on very different features. The problem seems to be that the underlying connections are very complex and non linear, meaning that a similarity measure would be, too.

---

### Official Review · Reviewer_UrY1 · 2024-11-03

**Soundness:** 3
**Presentation:** 3
**Contribution:** 2
**Rating:** 3
**Confidence:** 4

**Summary:**

The authors use GPT-4o to predict small molecules' ability to modify Magnesium's corrosion behavior based on a small dataset (<100 points). They ablate the performance of models with and without "contextual knowledge," which includes information such as the SMILES and some additional experimental details.

**Strengths:**

- The problem of predicting properties of materials in low-data regimes is relevant
- The authors report to beat a simple baseline
- The prompt for making the predictions seems novel in the field and relatively elaborate

**Weaknesses:**

- The authors report experiments on only one dataset with only one split (i.e., no error bars). It is unclear if the methodology transfers to other settings.
- The baseline is a simple neural network that has been previously reported. Other baselines (e.g., KNN - which mimics some of the suggestions in the prompt or finetuning the models similar to https://www.nature.com/articles/s42256-023-00788-1 or https://chemrxiv.org/engage/chemrxiv/article-details/668bd7385101a2ffa8c1e559)
- It is unclear how the authors arrived at the prompt and if there perhaps was some leakage while optimizing the prompt
- Only one LLM has been tested, and it is unclear how sensitive the performance is to the chosen LLM
The inclusion of context is similar to the concrete optimization example in https://pubs.rsc.org/en/content/articlelanding/2023/dd/d3dd00113j. In general, the methodology is relatively established.

The most important reason why this is not good for ICLR is that no novel technique has been reported. The others show that LLMs can, with a sophisticated prompt, perform some predictive tasks on one dataset. This is nice to see and adds to the existing literature on LLMs being able to perform such tasks. However, there is nothing novel that one can readily translate to another setting (at least the authors did not make clear what such a general methodology would be, and they also did not demonstrate transferability).

**Questions:**

- Why do you not perform greedy decoding (temperature = 0)?
- How did you arrive at the prompt? What was the optimization process? At which scores did you look for optimizing the prompt? How sensitive is the performance to the prompt?
- Would it be possible to perform an experiment to back up the claim "Analysis of the LLM’s outputs suggests that GPT-4o possesses more extensive knowledge about inhibition mechanisms than acceleration mechanisms, which may contribute to this outcome" could you construct similar datasets of inhibitors and accelerators and then perform an ablation?
- How has the parsing of the LLM output been performed? Was constrained decoding used? Was a regex used? Was it manually parsed?

---

> ### Author Response · Authors · 2024-11-18
> **Thank you for the constructive feedback!**
>
> Dear Reviewer,
> thank you very much for your review. I understand that our paper is not sufficient for the ICLR and therefore we will withdraw it. We thank you for your constructive feedback and your suggestions for improvement.
> Regarding your questions:
> - We tried greedy decoding with temperature almost 0 (that was not clearly stated, as "reducing the temperature..."), but this lead to significantly worse results.
> - We did manually write the prompt and optimized it manually in a couple of iterations on the training set. We tried some different prompts, where eg. a very simple prompt with just multishot prompting and without context enlarged the error by 50-100%.
> Analyzing the knowledge of GPT-4o with respect to inhibition and acceleration mechanisms is difficult, because:
> We do not know too much about the specific mechanisms ourselves
> The dataset is very small and sparse, especially on accelerator data (which probably also influences that fact)
> - We told the LLM in the prompt to write the number at last without anything after that and just read in the number from the end.

---

### Note · Authors · 2024-11-18

I have read and agree with the venue's withdrawal policy on behalf of myself and my co-authors.